# Granulomatous Secondary Syphilis: A Case Report with a Brief Overview of the Diagnostic Role of Immunohistochemistry

**DOI:** 10.3390/pathogens12081054

**Published:** 2023-08-18

**Authors:** Francesca Ambrogio, Gerardo Cazzato, Caterina Foti, Mauro Grandolfo, Gisella Biagina Mennuni, Gino Antonio Vena, Nicoletta Cassano, Teresa Lettini, Cosimo Castronovi, Vito Ingordo, Paolo Romita, Raffaele Filotico

**Affiliations:** 1Section of Dermatology and Venereology, Department of Precision and Regenerative Medicine and Ionian Area (DiMePRe-J), University of Bari “Aldo Moro”, 70124 Bari, Italy; dottambrogiofrancesca@gmail.com (F.A.); caterina.foti@uniba.it (C.F.); mauro.grandolfo@policlinico.ba.it (M.G.); gisellamennuni@gmail.com (G.B.M.); teresa.lettini@uniba.it (T.L.); cosimocastronovi90@gmail.com (C.C.); paolo.romita@uniba.it (P.R.); raffaele.filotico@uniba.it (R.F.); 2Section of Molecular Pathology, Department of Precision and Regenerative Medicine and Ionian Area (DiMePRe-J), University of Bari “Aldo Moro”, 70124 Bari, Italy; 3Dermatology and Venereology Private Practice, 76121 Barletta, Italy; ginovena@gmail.com (G.A.V.); nicoletta.cassano@yahoo.com (N.C.); 4Dermatology and Venereology Private Practice, 70125 Bari, Italy; 5Outpatients’ Department of Dermatology, District n.6, Local Health Centre Taranto, 74121 Taranto, Italy; vito.ingordo@gmail.com

**Keywords:** immunohistochemistry, granuloma, granulomatous histological pattern, secondary syphilis, *Treponema pallidum*

## Abstract

The diagnosis of syphilis can be challenging for dermatologists and dermatopathologists. In particular, secondary syphilis can have different clinical and histopathological presentations. A granulomatous tissue response is an unusual finding in secondary syphilis. We report the case of a 77-year-old man who presented with a 4-week history of non-pruritic generalised macules, papules, nodules and plaques. Histopathologically, there was a dense perivascular and periadnexal lympho-histiocytic dermal infiltrate with non-palisading and non-caseifying epithelioid granulomas and abundant plasma cells. The diagnosis of syphilis was confirmed by serology and immunohistochemical detection of *Treponema pallidum* in the biopsy specimen. A brief overview of the diagnostic role of immunohistochemistry is also provided, with particular emphasis on reported cases of granulomatous secondary syphilis.

## 1. Introduction

Syphilis is a sexually transmitted disease caused by the Gram-negative microaerophilic spirochete *Treponema pallidum*, which has been called ‘the great imitator’ due to its protean clinical features. Indeed, it can be mistaken for many other dermatological conditions and remains a diagnostic challenge for dermatologists and dermatopathologists [1,2,3,4]. Correct recognition of the disease and its appropriate treatment in the early stages are essential to prevent serious complications. The painless chancre of primary syphilis is often overlooked by patients. Many of them seek medical attention in the secondary phase of the disease, characterised by a rash that may be associated with non-specific systemic symptoms. The skin lesions of secondary syphilis may have different clinical and histological presentations that are sometimes atypical.

Granulomatous inflammation is an unusual finding in secondary syphilis and may represent a diagnostic pitfall, whereas a granulomatous response is commonly seen in tertiary syphilis [2]. Several hypotheses have been proposed to explain the development of a granulomatous response in secondary syphilis, such as correlation with disease duration and/or involvement of a hypersensitivity reaction to *Treponema pallidum* [3]. Different granulomatous patterns have been described in secondary syphilis, including poorly defined granulomatous inflammation, sarcoid and palisade configurations, interstitial granulomatous inflammation or tuberculoid granuloma [2,5]. Potential histopathological differential diagnoses in similar pictures include a wide variety of conditions, such as granuloma annulare, interstitial granulomatous dermatitis associated with systemic or drug-induced diseases, sarcoidosis, *Borrelia burgdorferi* infection, leprosy, leishmaniasis, tuberculosis, atypical mycobacterial infections and deep-seated fungal infections [2,5,6].

We report a case of granulomatous secondary syphilis in an elderly man, diagnosed using serology and immunohistochemistry for the detection of *Treponema pallidum* in the skin biopsy specimen. The purpose of our manuscript is to emphasise the importance of immunohistochemistry for the diagnosis of syphilis, especially for patients with atypical presentations, even if the diagnostic role of immunohistochemistry in syphilis is probably overlooked in dermatological practice. Indeed, we provide a brief overview of the diagnostic role of immunohistochemical detection of *Treponema pallidum* in skin samples, focusing on the reported cases of granulomatous secondary syphilis.

## 2. Case Presentation

A 77-year-old man presented with a 4-week history of a non-pruritic eruption of disseminated skin lesions. He denied the presence of weight loss, fever, malaise or other symptoms. The patient suffered only from arterial hypertension treated with telmisartan for many years.

Prior to referral, routine laboratory test results revealed no relevant abnormalities, and an immunohistopathological analysis of a skin biopsy sample showed a polymorphous cellular infiltrate consisting of T- and B-lymphocytes, in the absence of atypical CD30+ cells, with numerous polyclonal plasma cells and histiocytes and some multinucleated giant cells arranged in scattered granulomas. Treatment with oral prednisone was initially prescribed, but it produced no benefit. Therefore, the patient’s general practitioner and private dermatologist recommended further evaluation at our outpatient clinic.

Physical examination at the time of the visit revealed numerous red-brownish papules with some macular and nodular lesions on his head, neck, trunk and limbs (Figure 1 and Figure 2), and lesions that coalesced to form plaques on the face (Figure 2A) and legs. A few discrete lesions were observed on the palms (Figure 1B) and plantar surfaces, while the genital areas were spared. A minority of lesions had a squamous surface. Painless opaline plaques in the oral cavity (Figure 3) and micropolyadenopathy were also present.

Histopathological examination of a new skin biopsy specimen disclosed a dense perivascular and periadnexal lympho-histiocytic dermal infiltrate, with non-palisading epithelioid granulomas and abundant plasma cells (Figure 4, Figure 5 and Figure 6).

Ziehl–Neelsen and Grocott–Gomori’s methenamine silver stains were negative (not shown).

Additional laboratory investigations included serum angiotensin-converting enzyme, complement and autoantibodies. Antinuclear antibodies were positive at a titre of 1:320, and the other parameters were within the normal range. Moreover, the Venereal Disease Research Laboratory (VDRL) test and *Treponema pallidum* Haemagglutination Assay (TPHA) were positive, with titres of 1:32 and 1:640, respectively. Serology for hepatitis B and C viruses and human immunodeficiency virus (HIV) was non-reactive. The patient had no personal history of prior syphilis or other sexually transmitted diseases and denied any sexual activity over a long period of time, which remained unspecified despite repeated questions.

Taking into consideration the histopathological and serological findings, we decided to evaluate the presence of spirochetes in the biopsy specimen using immunohistochemistry for *Treponema pallidum* (primary polyclonal antibodies anti-*Treponema pallidum*, Biocare Medical, Concord, CA, USA, diluted at 1:400). Occasional spirochetes consistent with the *Treponema pallidum* species were detected (Figure 7).

Treatment with 2.4 million units of intramuscular benzathine penicillin once weekly for three weeks caused a progressive improvement in skin lesions that resolved after 2 months with slight dyschromic changes. There was also a complete resolution of oral lesions and lymphadenopathy.

The VDRL titre showed a fourfold decline after 6 months.

## 3. Discussion

Syphilis has caused immense human suffering throughout history; however, the discovery of penicillin made it easy to treat, and its incidence has decreased dramatically over time. Nevertheless, there has recently been a resurgence in the incidence of syphilis, particularly in association with HIV co-infection [1]. Serologic testing with the combination of treponemal and nontreponemal tests is mandatory for the diagnosis of syphilis. However, the interpretation of such investigations can be difficult and no single test has been shown to have sufficient sensitivity and specificity to recognise all stages of the disease [7]. Furthermore, false-negative serology and delayed seroreactivity have been described in patients co-infected with HIV [8,9]. The direct detection of *Treponema pallidum* in tissue sections may be useful to confirm the diagnosis, at least in particular circumstances and especially in individuals with atypical presentations and/or equivocal serological test results. It should be noted that numerous spirochetes can be found in skin lesions during primary and early secondary syphilis, whereas spirochetes tend to disappear in later stages of the disease [10]. Several methods have been developed for the detection of *Treponema pallidum* [7,8].

Silver staining techniques, such as Warthin–Starry staining, are traditional methods used to identify *Treponema pallidum* in formalin-fixed, paraffin-embedded tissue biopsies, mostly from patients with primary or secondary syphilis, but these methods have low sensitivity and may produce false-positive or false-negative results [7]. Immunohistochemical analysis with primary antibodies against *Treponema pallidum* is more sensitive than silver staining and is significantly more sensitive and specific than dark-field microscopy on a lesion exudate [7,10,11]. Compared with a clinical diagnosis of secondary syphilis, immunohistochemistry using the avidin–biotin–peroxidase complex method has demonstrated excellent specificity in all studies, with a sensitivity ranging from 49% to 92% [7].

In a study evaluating 40 skin biopsy samples from patients with syphilis (72.5% with secondary syphilis) and 36 skin samples from control subjects with other diseases [12], the Warthin–Starry technique did not accurately identify spirochetes in any sample, giving negative or inconclusive results, whereas immunohistochemistry detected spirochetes only in skin samples from patients with syphilis and was associated with a sensitivity of 60%, a specificity of 100% and an accuracy of 78.9%.

However, some authors emphasised that the interpretation of immunohistochemical results must take clinical data into account and recommended a rigorous search for the typical *Treponema pallidum* histomorphology due to possible false-positive results with alcohol-resistant bacilli (such as *Mycobacterium marinum*) and other spirochetes, including *Borrelia burgdorferi* and *Brachyspira* [5,13,14].

Data from patients with secondary syphilis have suggested the absence of significant correlations between the detection of spirochetes via immunohistochemistry and serological titres or certain histological features, such as the pattern of inflammation, the degree of infiltrate, the number of plasma cells and the presence of granulomas [11].

In addition, the identification of *Treponema pallidum* in tissue sections via immunohistochemistry may be useful to clarify the aetiology of a granulomatous dermatitis [5], as in our case.

The literature contains a limited number of reports mentioning the use of immunohistochemical detection of *Treponema pallidum* in skin samples characterised by granulomatous inflammation to support the diagnosis of syphilis. We performed a literature review of cases of granulomatous secondary syphilis in which the diagnostic work-up included immunohistochemistry for the detection of spirochetes in skin biopsy specimens. Table 1 summarises the main findings observed in these cases [2,3,5,6,15,16,17,18,19,20,21,22,23]. 

The literature search was performed in the PubMed database using the keywords ‘granuloma’ and ‘syphilis’ or ‘granulomatous syphilis’ and was limited to articles written in English and published until March 2023. In the reported cases of granulomatous secondary syphilis with available immunohistochemical results (Table 1), spirochetes in varying amounts were detected in most cases. In fact, immunohistochemistry did not demonstrate the presence of spirochetes in skin lesions in only one case [6].

As shown in Table 1 and previously described in a review article [2], granulomatous secondary syphilis can have a wide variety of clinical presentations. Papular and/or nodular eruptions have been most commonly observed, although a papular or nodular eruption does not necessarily correspond to a granulomatous histopathological picture. Skin lesions may vary in duration. It has been hypothesised that a granulomatous reaction occurs more frequently in secondary syphilis lesions with a long duration and has indeed been associated with manifestations lasting more than 4 weeks, although a granulomatous histopathological pattern has been documented in cases of shorter duration, even present for only 10 days before diagnosis [2]. As previously pointed out [2], involvement of the palmoplantar surfaces was absent in the majority of reported cases of granulomatous secondary syphilis and involvement of the oral mucosa was rarely described. Instead, our patient had some palmoplantar and oral lesions that contributed to the clinical suspicion of secondary syphilis. Another diagnostic clue for syphilis was the conspicuous infiltration of plasma cells on histopathology [4,24,25]. Despite positive serological test results for syphilis, our patient repeatedly reported a total absence of sexual intercourse for a long period time. We used immunohistochemistry with a polyclonal antibody against *Treponema pallidum* as an auxiliary diagnostic tool to assess the presence of spirochetes in lesional skin.

Our case confirms that the diagnosis of secondary syphilis must be considered in the context of granulomatous inflammation, especially in the presence of increased plasma cells [2], requiring further confirmatory tests. Serology remains of paramount importance, but immunohistochemistry may be regarded as a useful diagnostic tool, especially in patients with atypical or unclear features. Nevertheless, immunohistochemistry for *Treponema pallidum* is probably not very commonly used and known in dermatological practice.

In addition, syphilis should be included in the list of differential clinical diagnoses for diffuse eruptions of recent onset [23].

## Figures and Tables

**Figure 1 pathogens-12-01054-f001:**
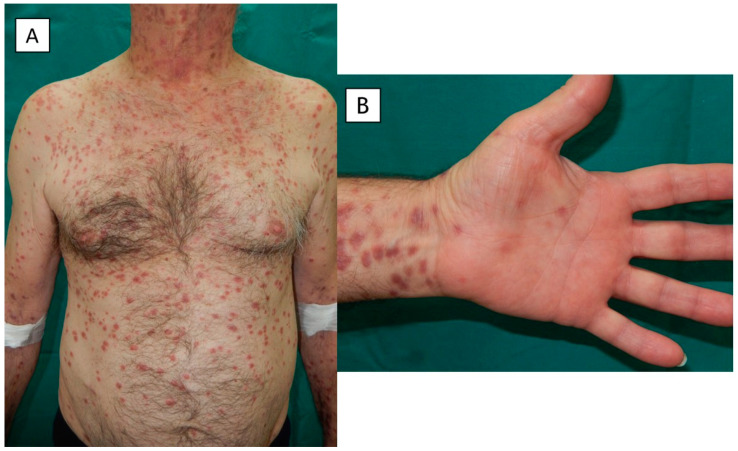
(**A**) Numerous brownish red papules with some macular and nodular lesions on the trunk and upper limbs, and (**B**) a few discrete lesions on the palmar surfaces.

**Figure 2 pathogens-12-01054-f002:**
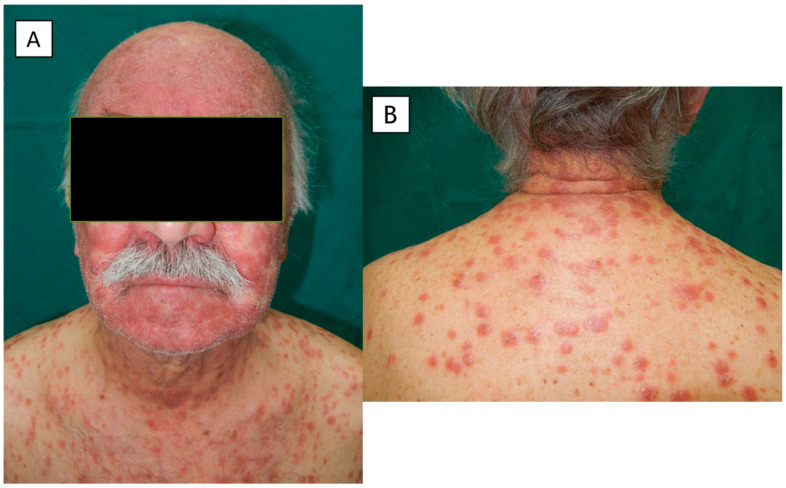
(**A**,**B**) Macules, papules and nodules on the face, neck and trunk.

**Figure 3 pathogens-12-01054-f003:**
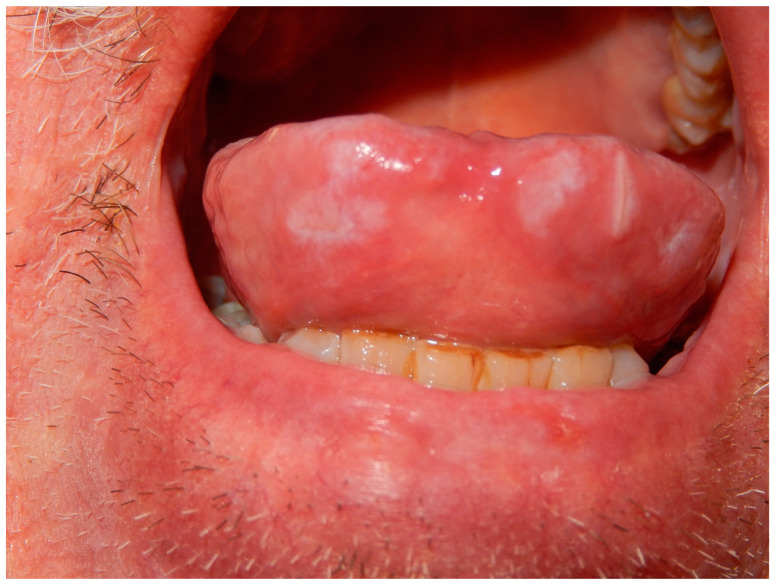
Opaline plaques in the oral cavity.

**Figure 4 pathogens-12-01054-f004:**
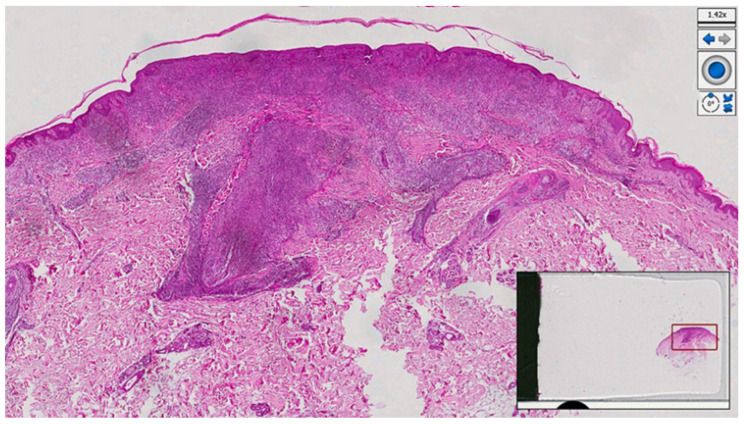
Histological photomicrograph showing a dense, robust, band-like inflammation underneath the epidermis, and brisk superficial and deep perivascular inflammation (haematoxylin–eosin, original magnification 4×).

**Figure 5 pathogens-12-01054-f005:**
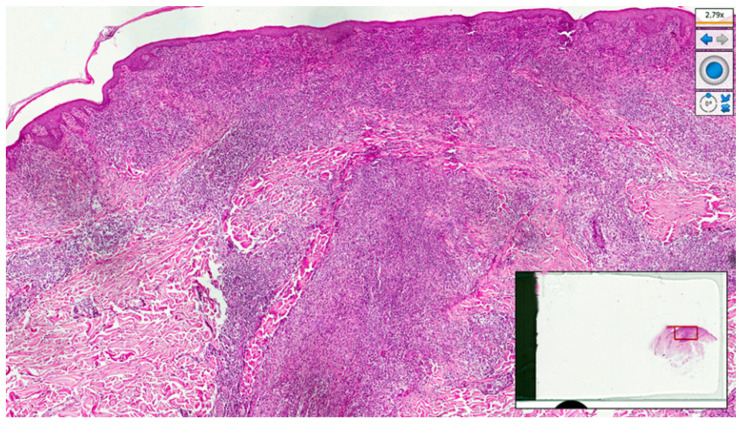
Periadnexal and perivascular lympho-plasmocytic and histiocytic infiltrate (haematoxylin–eosin, original magnification 10×).

**Figure 6 pathogens-12-01054-f006:**
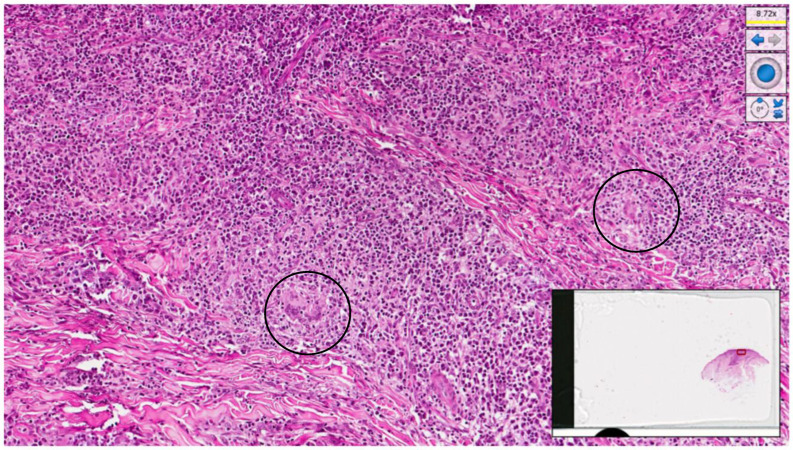
Histological micrograph showing, at higher magnification, a lympho-plasmocytic and granulomatous infiltrate with some giant cells (black circle) (haematoxylin–eosin, original magnification 20×).

**Figure 7 pathogens-12-01054-f007:**
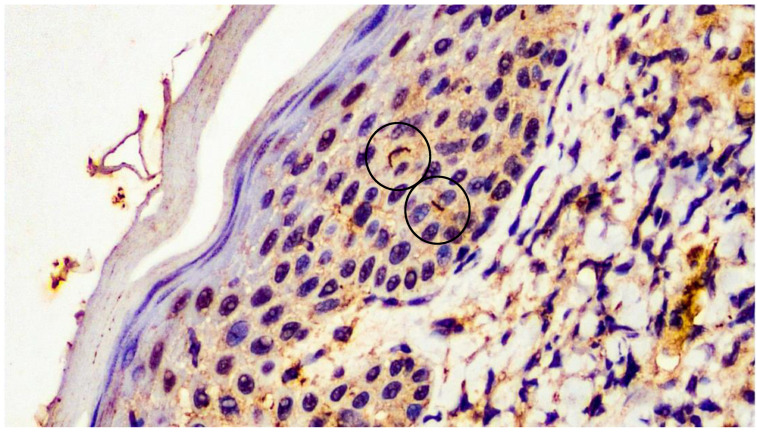
Immunohistochemical analysis with anti-*Treponema pallidum* monoclonal antibody: presence of filamentous structures/spirochetes in the epidermis (black circle) (immunohistochemistry for anti-*Treponema pallidum*, original magnification 20×).

**Table 1 pathogens-12-01054-t001:** Cases of granulomatous secondary syphilis present in the literature reporting results of immunohistochemistry for *Treponema pallidum*.

First Author [Reference]	Sex	Age (yrs)	Cutaneous Lesions (Duration, Type and Distribution)	Other Reported Clinical Information	Histopathology	IHC Results	Special Stains *
Sezer [5]	Male	47	3 months Non-pruritic macules and papules mostly on the trunk and extremities A few lesions on a palm.	-	Minimal acanthosis, focal basal vacuolar changes; rare dyskeratotic keratinocytes In the superficial and mid-to-deep dermis, moderate perivascular and interstitial infiltrate with numerous plasma cells, lymphocytes and a few neutrophils; histiocytes with occasional multinucleated giant cells (interstitial granuloma annulare-like pattern)	Positive (numerous spirochetes, mostly within endothelial cells and less frequently within macrophages or the basal epidermis)	-
Glatz [15]	Female	21	2 months Painless red patches and scaling, ulcerated, nodules on the face, neck and upper parts of the trunk and arms	No involvement of palms and soles No mucosal lesions No lymphadenopathy Headache No fever, malaise or weight loss	Hyperkeratosis, acanthosis and spongiosis of the epidermis with focal erosion; interface dermatitis Noncaseating granulomas in the whole dermis with multinucleated giant cells, interstitial and perivascular lymphohistiocytic dermal infiltrate, eosinophils and plasma cells	Positive (spirochetes in the dermis)	PAS, Brown-Brenn-Gram, ZN
Rysgaard [2]	Male	62	10 days Numerous nonpruritic red-brown, focally orange, papules and nodules on the trunk and upper extremities	No involvement of palms and soles No mucosal lesions	Marked perivascular and periadnexal lymphohistiocytic infiltrate in the dermis; non-palisading epithelioid granulomas with numerous plasma cells and multinucleated giant cells with elastophagocytosis	Positive (occasional spirochetes within granulomatous foci)	Gram, GMS, AFB, WS
Magdaleno-Tapial [3]	Male	55	30 days Multiple erythematous nodules and annular plaques on the face, neck, arms and upper parts of the trunk	No involvement of palms and soles No mucosal lesions No lymphadenopathy No systemic symptoms Alopecia in the occipital region	Psoriasiform epidermal hyperplasia Granulomatous infiltrate in the papillary and reticular dermis with epithelioid histiocytes and numerous multinucleated giant cells (tuberculoid granuloma), surrounded by a dense lymphoplasmacytic infiltrate; capillaries with edematous walls and prominent endothelial cells	Positive (especially in the epidermis and the epidermal ridges)	-
Hinojosa [16,17]	Female	25	8 weeks Hundreds of indurated erythematous itchy and painful papules on the face, neck, trunk and extremities (papulo-squamous on the trunk); some nodular or umbilicated lesions. A few brown palmoplantar macules	Dyspnea, malaise, myalgia (recent episode of fever, chills and sore throat) Genital lesions	Mild acanthosis, parakeratosis; neutrophils in the cornified layer Superficial and deep lymphohistiocytic infiltrate with periadnexal accentuation, epithelioid granulomas containing multinucleated giant cells and neutrophilic abscesses with a few plasma cells and eosinophils	Positive (multiple spirochetes in the epidermis, dermis and within granulomatous infiltrate)	Gram, PAS, Fite
Lee [18]	Male	29	2–3 weeks Asymptomatic, erythematous annular plaques only on the face	No involvement of palms and soles No mucosal or genital lesions No systemic symptoms No lymphadenopathy	Upper dermal perivascular and interstitial inflammatory infiltrates composed of lymphocytes and scattered plasma cells with a small naked granuloma	Positive	PAS; Alcian blue
Phan [19]	Female	30	4 weeks Erythematous papules with inward facing scale, primarily around the buttocks, trunk, and abdomen	No systemic symptoms Mild alterations of liver function tests Abdominal lymphadenopathy	Irregular acanthosis, mild spongiosis Granulomatous inflammation in the superficial dermis with epithelioid histiocytes forming granulomata, mostly around hair follicles and eccrine ducts; focal necrosis with several neutrophils within most granulomata; perivascular lymphohistiocytic infiltrate with occasional plasma cells and neutrophils in the superficial dermis	Positive (scattered elongated organisms)	PAS, GMS, ZN, Wade-Fite, PUTT
Henebeng [20]	Male	65	5 months Violaceous macules on the arms, legs, palms, and soles	At the onset, fever, rigors, and cervical lymphadenopathy No headache, weight changes, myalgia, numbness, or vision problems	Vacuolar interface alteration, focal exocytosis of lymphocytes Mild superficial and mild dermal perivascular lymphoplasmacytic inflammation; granuloma with interstitial giant cells and elastophagocytosis	Positive (rare intraepidermal spirochetes)	-
Fernández Camporro [21]	Female	51	15 days Mildly pruritic papules predominantly on the abdomen and top of the limbs	No involvement of palms and soles No fever or other systemic symptoms No genital or oral lesions No lymphadenopathy	Normal epidermis Predominantly perivascular nodular inflammatory infiltrate in the upper dermis comprising histiocytic cells, several multinucleated giant cells, epithelioid cells, lymphocytes, and abundant plasma cells, without caseous degeneration; endothelial edema and swelling	Positive (abundant intracellular and extracellular helical structures)	-
Jin [6]	Female	44	1 month Pruritic scattered erythematous, edematous papules and nodules on the extremities, trunk, and face	No involvement of palms and soles	Dense nodular lymphohistiocytic dermal infiltrate with periadnexal involvement; tuberculoid granulomas with numerous plasma cells	Negative	AFB, GMS, WS
Yousefian [22]	Male	20	6 weeks Multiple erythematous nodules on the forehead, nose, and upper limbs	Severe headache, photophobia, fatigue Cervical lymphadenopathy	Dense dermal granulomatous infiltrate with lymphocytes and numerous plasma cells	Positive (numerous spirochetes)	-
Saal [23]	Male	25	3 months Mildly pruritic edematous, grouped, and follicular papules mostly on the trunk and upper extremities A small, hyperpigmented patch on the heel of the left foot	No systemic symptoms Alopecic patches on the scalp	Mild lichenoid infiltrate and multiple non-caseating granulomas in the dermis; prominent infiltrate of lymphocytes and plasma cells	Positive (numerous spirochetes in the epidermis and dermis)	PAS, AFB

AFB, acid-fast bacillus; GMS, Gomori methenamine silver; IHC, immunohistochemistry; PAS, Periodic Acid-Schiff; WS, Warthin–Starry; ZN, Ziehl-Neelsen. * Negative results with the special stains listed in the last column in all reports.

## Data Availability

Not applicable.

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
