# Peer review of "Granulomatous Secondary Syphilis: A Case Report with a Brief Overview of the Diagnostic Role of Immunohistochemistry"

_pathogens, 2023, doi:10.3390/pathogens12081054_

Round 1

Reviewer 1 Report

The efforts are appreciated, however there are clear issues with the manuscript. Most importantly, there is minimal clinical relevance demonstrated by this case report, and it offers little to clinicians.

Note also, the introduction is far too short and Table 1 is not available in the main file. Regardless, the literature review should be much better integrated into the main findings - it seems that it was added here as little else than an afterthought. 

Fine, but reread for grammar

Author Response

Reviewer 1

Comments and Suggestions for Authors

The efforts are appreciated, however there are clear issues with the manuscript. Most importantly, there is minimal clinical relevance demonstrated by this case report, and it offers little to clinicians.

Note also, the introduction is far too short and Table 1 is not available in the main file.

Re: We added more information about granulomatous secondary syphilis in the introduction. Table 1 was not included in the main file but it was present in a separate file. Obviously, there was a problem in inserting the table. We are very sorry.

Regardless, the literature review should be much better integrated into the main findings - it seems that it was added here as little else than an afterthought. 

Re: Thank you; we tried to improve this issue.

Comments on the Quality of English Language Fine, but reread for grammar

Re: Thank you. We further checked the text for grammar.

Reviewer 2 Report

This case report explores granulomatous secondary syphilis and the diagnostic role of immunohistochemistry. The title properly reflects the subject of the paper. The topic is interesting; however, the manuscript is not well written. I have the following suggestions:

1. The writing and language are not clear. The standard of English used and the sentence structure needs improvement. The discussion part is difficult to follow. The entire discussion part needs to be rewritten and re-structured so that it makes more sense. Also, in the end, a good summary paragraph needs to be added.  

2. Add a high-power microscopic H&E picture, clearly showing histiocytic infiltrate/granulomatous inflammation.

3. Was Warthin-Starry stain done in this case? If yes, add the results and a picture.

4. In the discussion part, Table 1 is referenced. However, the manuscript available for my review did not contain the Table!

Thanks.

See comments to authors above.

Author Response

Reviewer 2

Comments and Suggestions for Authors

This case report explores granulomatous secondary syphilis and the diagnostic role of immunohistochemistry. The title properly reflects the subject of the paper. The topic is interesting; however, the manuscript is not well written. I have the following suggestions:

  1. The writing and language are not clear. The standard of English used and the sentence structure needs improvement. The discussion part is difficult to follow. The entire discussion part needs to be rewritten and re-structured so that it makes more sense. Also, in the end, a good summary paragraph needs to be added.  

Re: Thank you. You are right. Probably, the difficulties present in the discussion were mostly related to the different issues we examined and the great amount of data we summarized.  The discussion was rewritten, as suggested, hoping that it is clearer in the current version.

  1. Add a high-power microscopic H&E picture, clearly showing histiocytic infiltrate/granulomatous inflammation.

Re: A high-power microscopic H&E picture was added as Figure 6, substituting the previous one.

  1. Was Warthin-Starry stain done in this case? If yes, add the results and a picture.

Re: We are sorry. Warthin-Starry stain was not performed.

  1. In the discussion part, Table 1 is referenced. However, the manuscript available for my review did not contain the Table! Thanks.

Re: Thank you, we are very sorry but there were problems when inserting the table in a separate file.

Comments on the Quality of English Language See comments to authors above.

Reviewer 3 Report

The outpatient clinic that the autors work out should be described for the reader.

Advise who referred the patient to the authors clinic and who ordered the lab tests and conducted the biopsy

Table 1 is not seen in the manuscript-yet it is referenced?

The syphilis was staged as secondary syphilis-yet the treatment provided was for late latent syphilis?-Weekly benzathine penicillin over 3 weeks. Secondary syphilis treated as a stat dose of Benzathine penicillin 2.4 million units

The discussion is disorganised and difficult to follow and read. This should be re-done and the literature fully discussed in a methodical fashion. Comparsions to this case presentation and how it differs to the cases in the literature should be clearly delineated

The authors did not discuss the role of TPPCR (NAAT) testing of the oral lesions

When the patient last had sex was not indicated? "long period of time" is subjective

Please note in your case discussion how your case differs from the case referenced at 9 in your references

Minor grammatical errors only

Author Response

Reviewer 3

Comments and Suggestions for Authors

The outpatient clinic that the autors work out should be described for the reader.

Advise who referred the patient to the authors clinic and who ordered the lab tests and conducted the biopsy

Re: The patient was referred to our clinic by his GP and private dermatologist. This information was added.

Table 1 is not seen in the manuscript-yet it is referenced?

Re: Table 1 was inserted in a separate file but it was not present in the final document because of an unintentional error.

The syphilis was staged as secondary syphilis-yet the treatment provided was for late latent syphilis?-Weekly benzathine penicillin over 3 weeks. Secondary syphilis treated as a stat dose of Benzathine penicillin 2.4 million units

Re: You are perfectly right. The best treatment approach was repeatedly discussed and carefully considered. The presence of granulomatous inflammation on histopathology and the unknown time interval from sexual intercourses led us to be particularly prudent. Moreover, we took into account that treatment with three weekly doses of benzathine penicillin 2.4 MU was previously used by other authors in similar cases of granulomatous secondary syphilis (e.g., Glatz et al; Rysgaard et al; Lee et al; Fernandes Camporro et al; Jin et al).

The discussion is disorganised and difficult to follow and read. This should be re-done and the literature fully discussed in a methodical fashion. Comparsions to this case presentation and how it differs to the cases in the literature should be clearly delineated

Re: The discussion was re-written. The most relevant findings in our case, as reported in the discussion, were the presence of palmo-plantar and oral lesions that contributed to the suspicion for secondary syphilis. Instead, the review of literature revealed that the involvement of palmoplantar surfaces was absent in many cases of granulomatous secondary syphilis and the involvement of oral mucosa was rarely described. Moreover, an interesting finding was the old age of our patient.   

The authors did not discuss the role of TPPCR (NAAT) testing of the oral lesions.

Re: It was not discussed because it was not performed in our case as NAAT testing is not available in our hospital.

When the patient last had sex was not indicated? "long period of time" is subjective

Re: You are right, but the patient was particularly reluctant and did not provide any details despite repeated questions. This created further diagnostic difficulties.

Please note in your case discussion how your case differs from the case referenced at 9 in your references

Re: The case referenced at 9 was not characterized by a granulomatous histological pattern.

Comments on the Quality of English Language Minor grammatical errors only

Re: We checked the text for grammar.

Round 2

Reviewer 1 Report

The manuscript has improved, however the clinical relevance of this work has still not been adequately demonstrated. This needs to be made much more clear for this paper to have any real value.

n/a

Author Response

Reviewer 1

Comments and Suggestions for Authors

The manuscript has improved, however the clinical relevance of this work has still not been adequately demonstrated. This needs to be made much more clear for this paper to have any real value.

Re:We tried to explain better in the text the relevance of our work.We wanted to emphasize the importance of immunohistochemistry for the diagnosis of syphilis. It is a useful diagnostic method that is not frequently used for syphilis in clinical practice and not well known among dermatologists. It may be very useful in those cases with atypical presentations and/or unclear serology.

In addition, we performed a literature review focusing on cases of secondary granulomatous syphilis in which the diagnosis was supported by the immunohistochemical detection of spirochetes in skin biopsy samples and we summarized the characteristics of such cases (e.g., clinical and histopathological features and immunohistochemical results) in a table.

Comments on the Quality of English Languagen/a

Re: Thanks

Reviewer 2 Report

The prior concerns are answered by the authors appropriately. Please review the manuscript thoroughly again to improve language and make it more readable for the readers. 

The language can still be improved further.

Author Response

Dear Reviewer n'2,

thank you very much. We have corrected and improved english language of our manuscript. 

Thanks a lot

Reviewer 3 Report

The relevant changes have been appropriately corrected

This has been improved. a few spelling errors are noted

Author Response

Thank you very much.

We have improved the english language.

Round 3

Reviewer 1 Report

Sufficient for acceptance. Well done. Please re-read for any grammatical issues.

Author Response

Dear Reviewer n'1,

thank you very much for all. We have re-checked the manuscript for any grammar error.

Thanks again